# Exploring Community-Based Suicide Prevention in the Context of Rural Australia: A Qualitative Study

**DOI:** 10.3390/ijerph20032644

**Published:** 2023-02-01

**Authors:** Laura Grattidge, Ha Hoang, Jonathan Mond, David Lees, Denis Visentin, Stuart Auckland

**Affiliations:** 1Centre for Rural Health, University of Tasmania, Launceston 7250, Australia; 2School of Medicine, Western Sydney University, Penrith 2571, Australia; 3School of Nursing, University of Tasmania, Launceston 7250, Australia; 4School of Health Sciences, University of Tasmania, Launceston 7250, Australia

**Keywords:** suicide prevention, community-based suicide prevention, community development, regional and remote, mental health, program development, social capital, social innovation, positivist qualitative research

## Abstract

Suicide rates in rural communities are higher than in urban areas, and communities play a crucial role in suicide prevention. This study explores community-based suicide prevention using a qualitative research design. Semi-structured interviews and focus groups asked participants to explore community-based suicide prevention in the context of rural Australia. Participants recruited ((*n* = 37; ages 29–72, *Mean* = 46, *SD* = 9.56); female 62.2%; lived experience 48.6%) were self-identified experts, working in rural community-based suicide prevention (community services, program providers, research, and policy development) around Australia. Data were thematically analysed, identifying three themes relating to community-based suicide prevention: (i) Community led initiatives; (ii) Meeting community needs; and (iii) Programs to improve health and suicidality. Implementing community-based suicide prevention needs community-level engagement and partnerships, including with community leaders; gatekeepers; community members; people with lived experience; services; and professionals, to “get stuff done”. Available resources and social capital are utilised, with co-created interventions reflecting diverse lifestyles, beliefs, norms, and cultures. The definition of “community”, community needs, issues, and solutions need to be identified by communities themselves. Primarily non-clinical programs address determinants of health and suicidality and increase community awareness of suicide and its prevention, and the capacity to recognise and support people at risk. This study shows how community-based suicide prevention presents as a social innovation approach, seeing suicide as a social phenomenon, with community-based programs as the potential driver of social change, equipping communities with the “know how” to implement, monitor, and adjust community-based programs to fit community needs.

## 1. Introduction

Suicide is a major public health concern across communities worldwide. Additional risk in rural communities centres on social and geographic isolation, socioeconomic disadvantage and exposure to environmental disasters, such as climate change, floods, and bushfires, as well as lack of access to mental health services and poorer utilisation of existing services [1,2]. Such adversity leaves people in rural areas particularly vulnerable to mental health problems and suicide [3,4,5].

In Australia, as in other countries, age-standardised rates of suicide increase with remoteness, with rural men up to twice as likely to die by suicide compared to their urban counterparts [1,6]. In rural areas, suicide prevention programs need to be designed and delivered in ways that address these drivers of suicidality and best reach those within communities that are most at risk. Groups most at risk include people with mental ill health; people with a previous history of, or exposure to, suicide or suicidal behaviours; Aboriginal and Torres Strait Islander people; culturally and linguistically diverse people; lesbian, gay, bi, transsexual, intersex, queer/questioning and other gender and sexually diverse people; and younger people aged under 25 years, older people aged 50 and over, and men of all ages [1,7,8,9,10,11].

Across rural communities, broader social, environmental, and economic factors impact suicide, with social variables playing a prominent role in suicide, interacting with person-level factors to increase suicide risk. Stigma remains a key barrier to seeking help for all population groups, but particularly for men, compounding mental health problems and poorer quality of life [10,12,13]. In addition to stigma and the pressures of stoic ideals, which impact mental health service access in rural communities [10,14], social isolation is another suicide risk fact amplified by rurality. Social barriers and large distances therefore impact social connection and support, protective factors of mental ill health, and suicide [15,16,17]

Community-based programs are recognised as cost-effective approaches to suicide prevention, which can strengthen the motivation of participating regional partners within communities to take action, necessary for community ownerships and sustainability of efforts [8,18]. A recent report to the Australian Prime Minister on Suicide Prevention [19] suggested a number of ways to increase community capability to prevent suicide. Local-level efforts, delivered by community members, build on the premise that those who die by suicide in rural areas are likely to not have a diagnosed mental illness, nor have accessed formal mental health support leading up to death [12,13]. Suggestions made in this report emphasised a need for proactively using community as touchpoints for people at-risk to reach out to; increasing coordination of suicide prevention activities at a community-level; and developing new approaches to outreach activities and distress interventions. Therefore, to reduce suicidal behaviour and risk [19], public health and suicide prevention programs implemented in rural areas should be based on the needs of the local communities; the resources and facilities available; existing social connections and capital; and the broader environmental contexts and nuances of these local communities that impact suicidality and the effectiveness of suicide prevention efforts [18,20]. 

To date, suicide prevention programs specific to rural communities that have been independently evaluated and published have explored different methods of community engagement, building upon the diversity and cultural norms of rural communities, including incorporating community-led and participatory action research approaches [20,21,22,23,24]. Guidelines on how to undertake suicide prevention efforts at a rural community-level have also been published [1,18,25], including a report by the National Action Alliance for Suicide Prevention [26] in the United States, which, whilst not providing a definition of community-based suicide prevention, describes the necessary components of community-based initiatives. This report suggested that community-based efforts should be strategically planned using a unified model, where appropriate, of multiple, integrated suicide prevention strategies to fit, or align with, local context, culture, and readiness [26]. At an Australian level, the toolkit for suicide prevention in small rural communities published by Suicide Prevention Australia [18], whilst also not defining the term directly, provides guidelines for community-driven suicide prevention, informed by a rural Australian lens. When engaging community, terms like “partnerships” need to be defined within that specific context, to enable comparisons to be made and partners to be accountable [27,28]. When describing the partnerships needed for engagement in the community-based suicide prevention space, as previously described in this toolkit and for the purpose of this study, partnerships can include (without being limited to) community members (individuals and groups), community-based organisations and services, and formal and informal mental health and suicide prevention networks (incorporating either/both community members and professionals) [18]. Agents implementing community-based suicide prevention can be any of these roles present within the local community, or those planning or implementing programs at a broader policy level. Suicide prevention efforts, therefore, need to consider and account for the role of the agent, recognising common and overlapping areas of consideration, regardless of the type of intervention [29]. 

Suicide Prevention Australia released the Community Matter suicide prevention toolkit in 2014 [18]. Since then, various national programs have been implemented in Australia, including those delivered through the National Suicide Prevention Trial. Some of the programs implemented at the regional and rural community level through the Trial were evaluated, exploring the implementation processes and barriers to implementing national models at a regional level [30,31]. Previous evaluation of community-based efforts undertaken as part of the Tasmanian component of the National Suicide Prevention Trial between 2018 and 2021 [22,30,32] indicated poor and/or variable understanding of the operational definition of ‘community-based’ programs to be implemented under a systems-based model to suicide prevention. This proved a significant barrier to these efforts in that different stakeholders had different interpretations of the meaning of this term, leading to confusion and misunderstanding on what programs were appropriate to deliver at a local community-level, and how programs could be adapted to the local context. Program implementation of efforts under the National Suicide prevention Trial were further determined to be impacted by the roles of the agent, for example, the host organisation responsible for implementing the programs (for example Council, Neighbourhood House, service provider), levels of local community engagement, and resources available to build suicide prevention efforts. Community-level agents provided insight into how implementation of this national level program at a regional level relied on the same information to guide efforts, regardless of community characteristics or the role of the agent at the ground level [30,31]. Agents involved with suicide prevention in the field, therefore, have an insight into how programs can be implemented in rural communities, including thoughts on processes of engagement, and how to identify whether authentic learning, leadership, and empowerment have occurred within communities as a result of a program [28,33]. With this in mind, since the 2014 release of the toolkit [18] and the implementation of these national programs [30,31,34], no studies have sought to specifically explore community-based prevention in Australia and how implementation can be adjusted to fit local needs in the rural context. 

The current research question, ‘In context of rural areas, what are the key considerations when implementing community-based suicide prevention to meet community needs?’, proposes to explore how people define the term and account for variations in this definition when implementing programs. This research, therefore, seeks to explore developments over time in shifts in language, culture, or changes in the agents working in the field, from the perspectives of people with different experiences with implementing programs, research, and policies around Australia. Exploration of such perspectives reflects the role of the agents, and those factors impacting agents at the community-level, for example, how community consultation and the co-design process needs to differ, and how to enhance engagement with, and social capital of, those involved with programs. In addition, the role of agents at the policy level needs to be explored, for example, to increase the relevance of any guidelines published [28] and to best use local-level data to address any policy-implementation gaps, and ensure the suitability of community-based suicide prevention across settings in rural areas. This information can, therefore, help inform decisions on whether an intervention is appropriate for implementation or further feasibility testing, and what areas of methods or protocol need to be modified to suit the needs of specific communities [35]. 

## 2. Methods

### 2.1. Study Design

This study, specifically exploring community-based suicide prevention in rural areas, is the first phase of a larger study, exploring the role of rural communities in implementing youth suicide prevention, including the development of *‘Best Practice Guidelines for Youth Suicide Prevention in Rural Australian Communities’*. The present study focuses on the views from people with experience implementing programs in rural areas, which include community members, as well as people with a lived experience of suicide. The broader study will seek to explicitly include community members and people with lived experience of suicide living in rural areas.

For this study, a qualitative research design was utilised, with semi-structured interviews and focus groups with experts working in the field. Informed by a positivistic research approach, this qualitative design allowed for the perceptions of people in different roles of community-based suicide prevention to be examined with the aim to generate generalisations, focusing on the pure data without being influenced by human bias [36,37]. Interviews and focus groups have previously been used in suicide prevention research to generate insight and understanding around the complex factors contributing to suicide and impacting prevention and program implementation [22,30,38,39,40], including social and intercultural differences and recognising suicide and its prevention as no single viewpoint or issue [41].

Ethics approval was granted by the Human Research Ethics Committee (Project ID. 23582). The consolidated criteria for reporting qualitative research (COREQ), 32-item checklist for interviews and focus groups [42] was followed (Appendix A).

### 2.2. Participants

To ensure the recruitment of people with relevant expertise working in community-based suicide prevention, participants needed to have experience in paid positions working in this field, including being a community-level service provider, for example, health or community services; community-level or national-level suicide prevention program provider; or in a research or policy development role, for example, in academia working with rural communities, or with local, state, or federal government departments. Participants were purposively [43] recruited as potentially information-rich cases with varied perspectives and experiences relative to the phenomenon of interest [44], which in this study, is the knowledge of communities and their role in suicide prevention in rural areas. Inclusion criteria were, therefore, participants self-identifying as having expertise in rural suicide prevention (as part of the larger study for youth aged 12–25 years), aged over 18 years of age, and able to speak English. Participants were contacted using direct email invites to (i) contacts known to work in the area; (ii) as authors of relevant journal articles; and (iii) Google searches of people with these relevant experiences working in suicide prevention in Australia. Rather than seeking to explore the differences between participant groups, the current study sought to define a specific term (community-based suicide prevention), and therefore, explore similarities in thoughts on community-based suicide prevention across participant groups. A snowball recruitment method was also used as a means of enhancing the breadth and variation across the participant groups and whole sample [44], with participants asked to forward this invite to contacts working in the area, asking them to contact the research team if they wished to participate. This latter method helped reach people at the community level, as well as those working in the specific field of interest (community-based suicide prevention) [45]. Participants could be based anywhere in Australia (i.e., either rural or urban), provided they had some knowledge of rural suicide prevention or expertise implementing programs or working with, or living in, rural communities.

A total of 37 people participated in 32 individual interviews and 2 focus groups (*n* = 2, *n* = 3) (ages 29–72, *Mean* = 46, *SD* = 9.6; female 62.2%; lived experience 48.6%). Focus groups were contacts working in the same organisation or job role (in different locations), requesting the same session due to this connection/familiarity. The response options in the question relating to lived experience of suicide reflected Roses in the Ocean’s [46] definition, of “having experienced suicidal thoughts, survived a suicide attempt, cared for someone through suicidal crisis, or been bereaved by suicide”. Demographic characteristics of study participants are shown in Table 1.

### 2.3. Procedures

People were invited to participate in the study through email and provided a participant information sheet explaining the purpose of the study. Participants provided written, informed consent prior to interviews and focus groups. The ‘five phases of developing a semi-structured interview guide’ [47] was followed to develop the topic guide for interviews and focus groups. Each interview and focus group used the same semi-structured, open-ended questions (see Appendix A), including the open-ended question relevant to the present study, “how do you define community-based suicide prevention?” The study also elicited views on rural youth suicide prevention, the findings of which are being reported as part of the larger study, which is exploring the role of community in youth suicide prevention efforts in rural areas.

Interviews and focus groups were conducted between January and September 2021 via Zoom (*n* = 32) and face-to-face (*n* = 1) (COVID restrictions allowing) by the first author. This was considered appropriate given the low-risk nature of talking with professionals working in the field. It was made clear in the participant information sheet and in instructions discussed by the researcher prior to interviews and focus groups that a participant could stop the sessions and withdraw participation at any point. Given the sensitive nature of the topic and confidentiality issues with small groups, transcripts were not returned to participants for comment unless requested. No focus group or interview participants requested transcripts.

### 2.4. Data Analysis

Interviews and focus groups were continued until data saturation was reached [48]. Basic themes were identified after initial analysis and confirmed by a second researcher, and additional data did not lead to any new themes. Sessions were audio-recorded and transcribed verbatim by the first author, with data accuracy confirmed by a second researcher.

Data from participants’ transcripts were thematically analysed according to procedures suggested by Braun and Clarke [49], including initial thorough, systematic reading of each transcript separately, and then applying a reflective, inductive analysis process (including reading and re-reading transcripts) to search for meaning patterns. Transcripts were then coded separately by one researcher (the first author), confirmed by a second researcher, with themes being discussed with team members to reach consensus, allowing themes to emerge without forcing data into preconceived categories or theories [50]. 

## 3. Results

As can be seen in Figure 1, three overarching themes were identified from data analysis, relating to community-based suicide prevention within the context of rural areas: (1) Community led initiatives; (2) Meeting community needs; and (3) Programs to improve health and suicidality. These are thematically grouped and presented as a logical narrative as follows, along with supporting quotes, reflection on the context of each quote, discussion of subthemes identified, and any interplays between themes or participant groups.

### 3.1. Community-Led Initiatives

Participants spoke of community-based suicide prevention as grassroots approaches, with community involvement and community leadership at all stages of program design and implementation.

#### 3.1.1. Non-Clinical Efforts by Community Members and People with Lived Experience

Community-based suicide prevention are community-led efforts often delivered in non-clinical settings, engaging non-professional community members, including those with lived experience of suicide, to be involved in implementing programs.


*It’s equipping anyone who’s not a professional, anyone who’s not trained professional, especially as sort of a clinical professional.*

*Participant 18—Victoria, Service Provider*



*The simple one is non-clinical setting the community-based quite often also implies that it’s kind of community-led, or that there’s some strong engagement with people who are not necessarily professionals, in terms of some kind of service, but may have some other kind of expertise, some kind of lay, or lived expertise.*

*Participant 17—Northern Territory, Research*


The role of the community as program co-designers was also discussed, with the following participant involved with research seeing community involvement as providing important insight into how programs can best be developed and delivered at this community level.


*Local people that are living and based in that community, largely doing the work. But also, the design work, that they themselves, at the very start of doing this work, are involved in.*

*Participant 2—Western Australia, Research*


#### 3.1.2. Partnerships

Across participant groups, community coming together in roles that take action and implement programs was seen as a common component of community-based suicide prevention. Partnerships and working groups included agents in the field, including community members, local services, and professionals, and “natural supports”; those community members are naturally well placed to support those at risk of suicide and provide postvention support in the wake of a suicide.


*I would define it as a variety of natural supports, and professional supports, working together to provide everything from education to service support to ongoing care… I call natural supports everything from friends, family, sporting groups, church groups…*

*Participant 20, New South Wales, Suicide Prevention Program Provider*



*Disparate groups of people coming together to work because there’s been a suicide death in the community.*

*Participant 15—Tasmania, Policy*


In contrast, the following researchers in Victoria and the Northern Territory recognised the importance of partnerships and working groups to lead initiatives at a grassroots level, representing a broader range of community members, as well as those said to benefit from the program. This highlights opportunities for policy developers and researchers to not only understand and incorporate community needs into program development and implementation, but also how best to reach those at risk, by including those who may benefit from the program in program development. 


*It’s really a community led group… its proper grassroots community, they’ve got community members from all manner of groups represented, and services represented, as well as you just community members.*

*Participant 9—Victoria, Research*



*… it’s heavily involving people in the community who’s supposed to be benefiting from those programs.*

*Participant 17—Northern Territory, Research*


Partnering with local services was described as a key component of community-based suicide prevention, and from the perspective of someone working in the policy development/research space, this could allow for existing funding to be utilised to contribute to gaps in service access.


*Community and service providers coming and working together.*

*Participant 12—Tasmania, Service Provider*



*It’s really important to be able to work with the services who are funded, and where there is a gap.*

*Participant 16—Queensland, Policy/Research*


From the perspective of a community-based service and suicide prevention program providers in Tasmania and Western Australia working at the ground level, community-based suicide prevention was also seen as the way service providers and professionals work with community members. Working groups enabled the coming together of people as this partnership, the people that know their community best, collaborating formally and informally to meet the needs of those at risk in the community.


*I’ve seen that [community-based suicide prevention] work in action with the working group. To me, that’s mobilising community, that know the community…I’m getting goosebumps thinking about our working group.*

*Participant 9—Tasmania, Service Provider*



*A community working group… they’re usually made up of key, service, a delivery partners in the community… Health/Mental Health Services… the police emergency services… the best, most coordinated, responses… they operate both in and out of response. They’re meeting on a regular basis to share concerns that they might have around groups of people, and to collaborate to ensure that they’re working in a way that’s not, I guess doubling up…value-adding to one another in the work that they’re doing… when sadly there might be a death in that community, they’ve already got those relationships and those processes in place to enact really quickly in a way that’s already very considered.*

*Participant 1—Western Australia, Service and Suicide Prevention Program Provider*


#### 3.1.3. Natural Leaders and Gatekeepers

Community-based suicide prevention was described by the following service and suicide prevention program provider in Tasmania working at the community-level as efforts that engage natural leaders in the community, who “get stuff done”, the individuals and services implementing practical interventions at a grassroots level. 


*It’s about engagement and finding there’s always people in the community who get stuff done… if you look at community groups, there’s always a couple of people who are the key drivers…a core group of people who are there all the time, and they’re the ones who get a lot done… individuals, also with services…not to deliver a whole program… what are some practical interventions that they might utilise?*

*Participant 14—Tasmania, Service and Suicide Prevention Program Provider*


Community-based suicide prevention was discussed across participant groups as equipping the whole community to be informants on the lookout, people at the frontline recognising and linking those at risk with gatekeepers in the community, as well as professional supports. This recognised that people in the community do not always reach out to formal services and may rather turn to other members of the community for guidance or support. It also clarifies the need for increased community awareness and skills training, as well as community engagement, with professionals and community members needing to work together to help those at risk. 


*The focus on gatekeeper training and making sure that the community is as suicide aware as possible, so I guess that’s our best frontline defence…*

*Participant 27—New South Wales, Research*



*…equipping the general community is so important…they’re much more likely to have that risk in community allocated to them… up to 70 to 90% of people who are at risk of suicide or who died by suicide communicate to family, only 20 to 30% reach out to a professional.*

*Participant 18—Victoria, Service Provider*


From a service and suicide prevention program perspective, gatekeepers were seen as essential in recognising people at risk of suicide, having conversations safely, and providing support to people at risk before reaching crisis point. Community members implementing programs in these gatekeeper roles were seen as the frontline, much like a paramedic during a medical crisis—there at the right time, with the right tools and basic first aid. 


*…there needs to be community gatekeepers…people should have a better understanding of the risks to suicide, and how to support somebody, who to refer them to…*

*Participant 19—Tasmanian, Service Provider*



*Getting someone into the service before they’re a crisis point…So that’s the community’s role and part of the puzzle…if you talk to a lot of the paramedics… they’ve got all the great tools, they can do a lot of amazing stuff. But if the person isn’t there, right at the point of the accident happening, doing some simple basic stuff, that it doesn’t matter…*

*Participant 7—Queensland, Suicide Prevention Program Provider*


Community-based suicide prevention relies on natural leaders and gatekeepers in the community to be involved and “get stuff done”. Through partnering with existing services and professional support, community members delivered primarily non-clinical interventions and raised awareness, implementing programs based on local needs. 

### 3.2. Meeting Community Needs

In addition to discussing the role of the agent in implementing programs, community-based suicide prevention was also seen as determining, and being ideally responsive to, the needs identified by the communities themselves, addressing local issues and empowering communities to generate solutions to these issues. 

#### 3.2.1. Knowing “Community” 

Understanding who “community” is was a key consideration for implementing community-based suicide prevention. In one form, from the perspective of someone working in policy development, community refers to a group of people as regionally confined within a geographical area or as multiple communities within an area. 


*I think in terms of the regional or the geographical definition, it’s suicide prevention that is developed from and implemented in local communities or a specific geographical region.*

*Participant 21—New South Wales, Policy*


In contrast, from the perspective from someone working at service/suicide prevention provider level, community was also considered as a social construct, as diverse groups of people coming together for a common purpose or interest, across settings. Flexible membership arrangements encourage community engagement and allow a range of people to join a community group or network, meaning perspectives of group members can be shared and interconnections across settings established and maintained.


*It’s a group of people with a similar interest; workplace, men sheds, footy clubs, netball clubs…it’s about people who come together for some purpose…particularly sporting clubs, and community groups, where you get lots of divergent people coming together. golf clubs are a good example of that, where you have people from all ages, all walks of life. Their interest is golf, but in terms of what they do for their day job and stuff outside, it’s so diverse. So, when you engage with them, what it means is that you’re not just sort of getting a particular cohort, you’re getting on a really diverse group of people…*

*Participant 14—Tasmania, Service and Suicide Prevention Program Provider*


Seeing it from a broader perspective, the following researcher described community as a fusion of rurality, that is, the shared values, lifestyles, and norms that are experienced by people in the community on a daily basis, along with the intersection of culture.


*… it’s a mix of rural life and rurality if you like, and rural culture…It’s a coming together and fusion of those things.*

*Participant 13—South Australia, Research*


Community-based suicide prevention requires a shared understanding within the community involved in terms of perceptions and beliefs of what community needs are and how to address suicide in the community, and potential solutions to identified issues, utilising both formal and informal engagement strategies to work with local communities. This was understood by the following researcher to be particularly important within cultural contexts, such as Aboriginal and Torres Strait Islander communities, recognising the general community as an informant of knowledge, a resource to build on, and an important setting to implement programs―all of which were described as being demonstrated through the National Suicide Prevention Trial.


*There are some gaps. There’s this idea of the community level type response that, I think has always been there when it comes to suicide prevention, Aboriginal and Torres Strait Islander communities… the general population that’s increasingly becoming recognised as an important resource or site, for intervention…the community needs to have this kind of shared understanding, awareness, buy in, engagement, formal or informal, into whatever’s going on, in terms of the suicide prevention activities… certainly felt like I saw that pop up more in really tangible ways in the National Suicide Prevention Trials.*

*Participant 17—Northern Territory, Research*


Working with communities at the ground level, the following service/suicide prevention provider described how communities themselves, whether a group defined by geography or social connection, need to define who they are, recognising the importance of shared beliefs, local knowledge, lifestyles, and cultures.


*[Community is] however they define themselves, really.*

*Participant 14—Tasmania, Service and Suicide Prevention Program Provider*


#### 3.2.2. Identifying Needs

The following participant working in research/policy development found through their work that “success” was found when communities were involved in the co-creation of community-based suicide prevention programs to address local issues as self-identified by the community, including healing components for the whole of the community.


*It’s really about the community self-identifying what the issues are, and what would the best possible solutions be. That’s where I see the most success… when they have the opportunity to have those healing components included in their journey, is really important…*

*Participant 16—Queensland, Policy/Research*


From the perspective of a service provider working in the Northern Territory who worked from a community development framework, community needs should be understood through a community development lens, having regular conversations and ensuring current needs are addressed.


*I work from a framework of community development. So, you can’t really do anything in this area of work unless you tap into the community… community-based suicide prevention is about having key people on board that are constantly, regularly having conversations about need.*

*Participant 29—Northern Territory, Service Provider*


#### 3.2.3. Diversity and Cultures

At a community-level service-provision level, the following participant in New South Wales saw flexibility as being needed when working with community to determine what their cultural needs are and what community-based suicide prevention programs should be offered. 


*It needs to be very inclusive…every community will have its own culture; therefore it needs to be varied enough to be flexible to meet the needs of that community.*

*Participant 20—New South Wales, Service Provider*


The following participant involved with research/policy development found that through community informing and leading interventions, community-based suicide prevention can enhance community protective factors, community-wide healing, and restoration within Indigenous communities. By communities running interventions, impacts of these efforts were, therefore, seen as potentially amplified. 


*You get the good advice about what programs are suitable, but you get that additional protective factor…it’s restoration of things that have been taken away from indigenous people. And that in itself is empowering, that in itself is protective. I think that it’s a hard thing to get your head around sometimes. But it does mean that if you’ve got a successful program in place, then the effectiveness of it is amplified, because it’s not just the intervention, it’s the fact that the community is running the intervention.*

*Participant 4—Queensland, Policy/Research*


Community-based suicide prevention addresses the needs of communities, as identified by the communities themselves, empowering diverse groups within the local population to come together and take action. Understanding who the community is includes taking into account community characteristics and nuances, history and trauma, and the diverse and cultural needs within a community. 

### 3.3. Programs to Improve Health and Suicidality

To meet the needs of communities, researchers in the field further described how community-based suicide prevention is commonly delivered at a community-level, often in non-clinical settings, aiming to improve health and suicidality through addressing impacting factors. 

#### 3.3.1. Drivers of Suicide 

From the perspective of this participant involved in research in South Australia, the drivers of suicide and suicidal behaviour were seen as often being addressed through community-based suicide prevention in non-medical settings.


*Community-based suicide prevention is fundamentally about a thoughtfulness for the drivers and conditions that may lead to the onset and worsening of suicidal ideation or suicidal behaviour more broadly, and community-based suicide prevention is about the everyday understanding and wanting to do something in that context, which is unlike what would typically happen in a medical or other type of setting.*

*Participant 13—South Australia, Research*


The same researcher described how community-based suicide prevention also considers the broader social environment and living circumstances impacting on suicide risk, including the wider community setting and the role of stigma.


*Community-based suicide prevention is dependent upon understanding how stigma plays a part in the lives of people who seek help for suicide… understanding living circumstances, what exists in a community setting…*

*Participant 13—South Australia, Research*


As described by this researcher in Victoria, universal community-based suicide prevention programs build on the ongoing, intergenerational nature of communities, as well as community needs and characteristics. Through addressing whole populations at risk, programs, therefore, need to account for local culture, history, and the availability of resources. Universal programs implemented in rural communities can improve mental health and wellbeing, and awareness of suicide and its prevention, building resilience and protective factors across generations.


*Universal prevention approaches… resilience type programs or mental health awareness type programs that get rolled out across communities with the view that they would reduce suicide risk further down the track, if people are made to be terribly resilient when they’re kind of four or five years old.*

*Participant 9—Victoria, Researcher*


#### 3.3.2. Health, Wellbeing, and Quality of Life

Participants in policy development and research roles in Victoria and Tasmania saw community-based suicide prevention as those varied efforts that support the goals of community wellbeing and health, including those focusing on the social determinants of health.


*Support across the continuum for wellbeing and suicide prevention can be regarded as kin to community-based suicide prevention.*

*Participant 9—Victoria, Research*



*Anything to do with addressing social determinants of health is guide is, you know, supports improvement of health and wellbeing.*

*Participant 15—Tasmania, Policy*


The same participant working in a policy development role described how community-based suicide prevention needs community engagement and coordinated efforts delivered by specific services and local professionals, wrapping around and supporting community members to improve mental health and wellbeing.


*… when it gets down to community-based suicide prevention, I would probably start to drill right down into the specific services…community based mental health service, but it’s probably more likely to be your primary care and psychologists and then those that also deliver training, education and training and interventions in workforces… it’s a whole range of things. But to me anything that wraps around someone to improve mental health and wellbeing and suicide prevention broadly.*

*Participant 15—Tasmania, Policy*


Through community-based suicide prevention, community members come together as networks, supporting and improving the quality of life of fellow community members.


*There’s a genuine need to fully understand and lean in as a community person, or lean in as a community network, into the lives of somebody in that community so that life can be better, or circumstances can be better.*

*Participant 13—South Australia, Research*


#### 3.3.3. Varied Approaches, Mobilising Available Resources

A participant with experience as a service provider involved with a working group, outlined how community-based suicide prevention mobilises community to implement suicide prevention programs. Through their perspective, working groups were, therefore, essential for community members and service providers to come together and build on, and share, resources and social capital.


*I’m getting goosebumps thinking about our working group….it’s that community and service providers coming and working together, to resources to make sure there’s conversations.*

*Participant 9—Tasmania, Service Provider*


In contrast, this suicide prevention program provider saw the variability in the approaches that could be provided under the umbrella term, which was dependent on accessible services and supports, costs, and the needs of the target group at risk.


*…it would be a mixture of free, subsidized and at cost, because it would depend on who was aimed towards. It would be also offering a variety of methods. A mixture of free, subsidised and at cost, because it would depend on who was aimed towards…everything from face to face to telehealth to text and phone…*

*Participant 20, New South Wales, Suicide Prevention Program Provider*


The following participant working in a policy development role in Tasmania described community-based suicide prevention as those informal events or activities undertaken as community action, rather than just action plans.


*Your community might be, I don’t know, the CWA [Country Women’s Association], and you decide you’re going to say, have a sausage sizzle, cook some scones. Do this, do that, discuss that, do a bit of training and develop community action rather than a community action plan. That’s community-based suicide prevention to me.*

*Participant 15—Tasmania, Policy*


Service providers in Victoria and Tasmania described how community-based suicide prevention facilitates “getting the message out there”, through communities sharing information and increasing awareness around how to talk about suicide and its prevention safely.


*…it’s about getting the message out there.*

*Participant 31—Victoria, Service Provider*



*Community based suicide prevention is about everyone in the community, understanding what to say, and understanding how to have those conversations safely.*

*Participant 10—Tasmania, Suicide Prevention Program Provider*


The following suicide prevention program provider recognised that in addition to awareness raising, community-based suicide prevention are those programs implemented by community to provide coordinated, postvention responses.


*They run kind of community awareness stuff, but they also have a group that’s supports those that are bereaved by suicide.*

*Participant 34—Victoria, Suicide Prevention Program Provider*


Community-based suicide prevention is, therefore, referred to as the varied programs implemented by communities to address determinants of health and suicidality, often social in nature and impacting quality of life. Communities often implement these programs utilising available resources, implemented to the whole of communities, including to increase awareness and the capacity of communities to recognise and support those most at risk.

## 4. Discussion

This study explored community-based suicide prevention in the context of rural Australian communities, from the perspectives of experts currently working in the field. The findings provide direction for policy and national guidelines and potential areas to be further explored, including the effectiveness of approaches used under the umbrella term “community-based suicide prevention”, as described throughout, as well as its role as a driver for social innovation and change. 

### 4.1. Defining Community and Its Role

Throughout each theme, the various roles of community in community-based suicide prevention were described, ranging from implementing, co-design, collaborating and informing program planning and delivery. Participants from policy and researcher groups suggested community as being groups of people within the same geographical region, as well as those groups of people with shared values, lifestyles, and norms, who, at the ground level, can provide a more nuanced perspective of community-based suicide prevention. They further described community from a social perspective, the informal and formal groups that form with a common purpose. Across participant groups, community were described as being ideally placed to lead the implementation of initiatives; however, where this is not possible, community still have a role in co-designing and collaborating, and informing the development and delivery of programs. The importance in allowing the community to define who they are and what their role is, rather than this being imposed, is supported by the literature [18,26,28,51] on community-based suicide prevention, engagement, and co-design, both empowering the community involved and ensuring the relevance of the program. 

### 4.2. Driving Social Change

Building on the literature, this study reinforces that community-based suicide prevention efforts in rural areas are those primarily non-clinical efforts undertaken in community settings [18,20]. With the goal of improving community wellbeing, community-based suicide prevention presents as a social innovation approach, seeing suicide as a community-level social phenomenon, with community-based programs as the potential driver of social change [52]. Through this approach, communities are equipped with the knowledge of their communities, and how to implement suicide prevention initiatives to increase awareness, support those at risk, build protective factors, and reduce risk factors [1,2,19]. There is a continuous need to centre suicide prevention efforts on impacting social drivers of suicide. This includes enhancing protective factors, increasing community awareness, reducing stigma, and challenging stoic ideals, as well as providing opportunities to increase social connection and reduce isolation and impeding feelings of helplessness, and access to services and supports, all particularly prominent in rural communities [10,12,13,14,15,16,17]. 

### 4.3. Programs Improving Community Wellbeing and Other Protective Factors

The study findings purport that community wellbeing, and other secondary program outcomes, such as the building of protective factors, are important components of community-based suicide prevention, previously found to improve relevant social determinants of health, as well as reduce rates of suicidality [1,53]. The findings further emphasise the need for community-based programs to consider and accommodate for the diversity of lifestyles, local cultures, heritage and traditions, shared beliefs, values, and norms within communities, providing a holistic lens to suicide prevention and whole of community healing. This is particularly important for communities in rural areas who are impacted by suicide and trauma, including Aboriginal and Torres Strait Islander communities, with community-based suicide prevention providing a means for rural communities to address past trauma. This includes by providing opportunities for empowerment through being inclusive in decision making and the co-creation of programs, and connection, building social capital and community cohesion, which can also enhance community ownership and the sustainability of programs [54,55].

### 4.4. Building on Social Capital: Community Gatekeepers and Touchpoints

Where existing research and models stress the importance of frontline workers, such as first responders, to identify and support people at risk in community in rural areas [34,56], the present findings provide an additional perspective. Community members are at the very forefront, informants in the community on the lookout, providing crucial engagement and suicide “first aid”, identifying someone at risk, supporting them where possible, and linking them to appropriate supports. Participant groups all spoke of community-based approaches as being those efforts that increase the skills and confidence of community members to act as gatekeepers, recognising and supporting people at crisis point. These approaches have been shown by the research as essential for suicide prevention in rural areas, to utilise the social capital in communities, as a resource to build on in areas where resources are commonly scarce [57,58]. In addition, considering the environment as an impacting factor for suicide, community being touchpoints ensures safe spaces for people at risk to reach out to. This not only supports previous research [59], but also recommendations by the Advisor on Suicide Prevention to the Prime Minister, that a more connected and compassionate approach is needed to assist people at risk of suicide, taking help to the people where they are, and in a way that suits their needs [19].

### 4.5. The Need for Community Engagement and Partnerships

The study findings build on the recommendations made by the Australian Government [51] in 2005, that community-wide approaches can foster community ownership, whole of community engagement, and participation, with community as the local agents best placed to partner with and implement programs. Implementing community-based suicide prevention programs in rural areas relies on engaging trusted individuals and organisations in the community, including community leaders or champions who have knowledge of the needs of their community, and how programs can be best developed and delivered. Partnerships between community members, including people with lived experience of suicide, and local services and professionals, provide avenues for the exchange of knowledge, building the social capital of local communities. As described by several community-based service and suicide prevention program providers participating in this study, locally based working groups and formal and informal suicide prevention networks are crucial partnerships needed to engage a diverse range of community members to come together with the sole intent of planning and taking action against suicide. Having extensive experience working across communities, researchers participating in this study also recognised how these partnerships needed to include community members and people with lived experience of suicide at all stages of program planning and implementation. They described these processes as essential to ensure that programs are attuned to community needs [9,60], are relevant, and able to reach the people who need it most―recognised as pivotal factors in suicide prevention [19]. The present study findings support that community ownership of programs is enhanced through communities being involved in the co-design process, informing program planning and implementation [28]. Prioritising community engagement and inclusive partnerships is essential in these communities, where resources are often scarce, building on existing social capital and resources, to embed and sustain programs within everyday community life. 

### 4.6. Transitioning from National to Local-Level Approaches

This study highlights the ever-evolving and variable nature of both community and community-based suicide prevention. The roles of government tiers in supporting community-based efforts at regional and a national level, therefore, needs to be continually explored, along with ongoing assessment of how best to determine and meet community needs. This is particularly relevant in regional and rural areas of Australia, where, for example, the National Suicide Prevention Trial and place-based trials in Australia, have recently been implemented, highlighting the need to explore community-level factors, and community-based suicide prevention within the transition of suicide prevention efforts from a national to a regional level [19,22,30,32]. The findings not only suggest that future planning of suicide prevention efforts in rural areas of Australia need to engage communities and include them in the co-design processes, but that communities can boost the sustainability of efforts by owning and playing a key role in leading these efforts.

### 4.7. Study Limitations

The study limitations centre on variability regarding the number of participants from each state and demographic groups. Additional information on the participants providing quotes, for example, state/territory of residence and role, are provided as confidentiality allows and the person cannot be recognised. The participant groups, as representations of the wider social context of the people within them, may also reflect the differences in experiences working in the field. While these comparisons have been noted in the results and in the discussion, for example, from the perspective or research/policy versus service/program provider, there may be areas where these differing levels of power relations/experience may have influenced the information provided. Those in research and policy roles also reflected on past experiences working in service or program provider roles. Although participants reflected on their experiences from living and working across Australia, the study sample provides these participant’s views only, which may not be representative of, or generalisable to, the whole population of people working in the field or across all communities or rural areas. 

## 5. Conclusions

Transitioning suicide prevention efforts from a national to a regional focus requires empowering rural communities to draw on existing resources and social capital in order to implement suicide prevention programs supporting their own populations. This study explores community-based suicide prevention as published in existing guidelines [18,26], and from experts who work in the field, with consensus centring on a need to recognise, acknowledge, and support rural community organisations and members, including people with lived experience, to be involved with, design, and implement programs. As ground level experts, these people have an awareness of their community, including community needs and those most at risk. As previously highlighted, community-level implementation of programs requires communities to be able to understand what programs are most suitable to meet the needs of their communities, and how these can be best adapted.

The agents implementing suicide prevention in rural areas, whether professionals, gatekeepers and people at the frontline of community, other relevant non-professionals, researchers, policy makers, and program funders, therefore, need to remain aware of how prevention across geographical regions should be relevant to the population of the region, whether based on geography or social factors including the shared cultures, lifestyles, beliefs, and norms of groups of people in the community. Implemented with this in mind, community-based suicide prevention provides options to reach those most at risk in rural areas who are not, or cannot, access formal support services, addressing social drivers of suicide, including isolation and stigma, and building the community engagement needed to mobilise people to take action. To plan and implement community-based suicide prevention, guidance and guidelines are needed for both community-level agents and those working at a policy level, on how best to understand community needs, engage the community, and reach people most at risk, across all settings and sectors. This information can be used to inform what programs should be implemented and how these can be modified to meet local community needs. 

## Figures and Tables

**Figure 1 ijerph-20-02644-f001:**
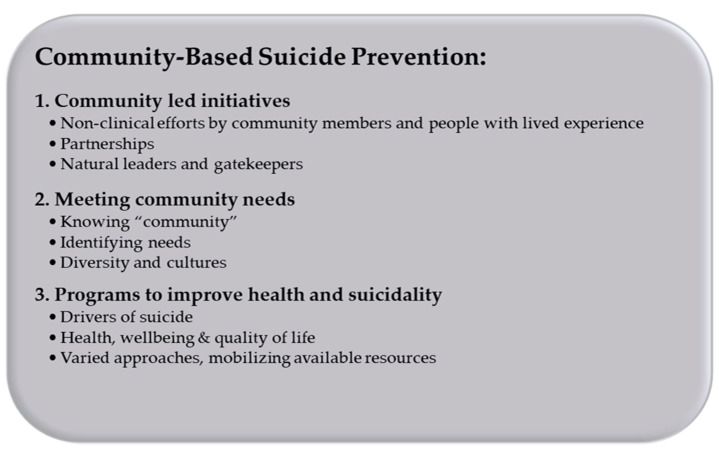
Thematic Groups and Subthemes Identified from Interview and Focus Group Data.

**Table 1 ijerph-20-02644-t001:** Participant demographic characteristics.

**Age** * (*n* = 35), range = 29–72, ***MEAN*** = 46 (*SD* = 9.6)		
**Gender**	** *N* **	** *%* **
Female	23	*62.2*
Male	11	*29.7*
Other (non-binary)	3	*8.1*
**State**		
South Australia	1	*2.7*
Northern territory	2	*5.4*
Victoria	4	*10.8*
Western Australia	6	*16.2*
New South Wales	7	*18.9*
Queensland	8	*21.6*
Tasmania	9	*24.3*
**Lived experience**		
Yes	20	*48.6*
No	9	*24.3*
Prefer not to say	6	*16.2*
Non response	2	*5.4*
**Aboriginal and/or Torres Strait Islander**		
Aboriginal	2	*5.6*
Neither Aboriginal nor Torres Strait Islander	34	*91.9*
Non response	1	*2.7*
**Role in community-based suicide prevention**		
Community-level service provider	18	*48.6*
Suicide prevention program provider	7	*18.9*
Policy or research	12	*32.5*

Abbreviations: *n*, number; *SD*, Standard deviation; * Missing data for age for two participants.

## Data Availability

Not applicable.

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
