# Peer review of "Exploring Community-Based Suicide Prevention in the Context of Rural Australia: A Qualitative Study"

_ijerph, 2023, doi:10.3390/ijerph20032644_

Round 1

Reviewer 1 Report

Nicely presented qualitative study that demonstrated rigor by the researchers.

1. How Community-Based Suicide Prevention is defined and used to reach individuals at risk in rural areas.

2. I think it is a significant topic and very relevant in the field of suicide and suicide prevention especially addressing rural geographical areas. Although this is a qualitative study and the results cannot be generalized, professionals, gatekeepers,  other relevant nonprofessional persons, as well as policy makers and funders of suicide prevention need to remain cognizant that prevention across geographical regions needs to be relevant to the population of the region.

3. It does add consideration for different geographical regions.  However, it is well-known or considered best practice, to involve the community members, including those with past experience into the study. In that sense, originality was not the strongest point of the study.  Another important point was inclusivity and meeting the needs and challenges of culturally diverse populations. Having researchers and policy personnel involved was addition to what I believe to be the usual participants, but I do wonder how their participation affected the study results.

4. The qualitative process appears to be sound with specific emerging themes.   I am curious about the researcher/policy participants.  They did participate in a rather large number and quite a few quotes from “researcher or policy” participants.

5. The tables and graphs were helpful when reading the emerging themes and qualitative statements.

Reviewer 2 Report

Grattidge et al, Defining Community-Based Suicide Prevention in the Context of Rural Australia: A Qualitative Study

The paper is about the community based suicide prevention practices. By means of an empirical qualitative study, it tries to come up with common characteristics of such interventions.  This is useful information to know how agents involved in the field look at this phenomenon. However it is not well stated why this information is important?? Is it about conflicting visions between policy and agent involved in the local implementation?

Main comments

So, the research problem is not formulated well; Why is it important to know what people in the domain of suicide prevention regard as community based programs?? You could state that literature about community engagement/community ownership as well as social innovations provide a framework to define the concept and to study it in the field. In my opinion, the concept is not defined in the paper, but the common characteristics of community based suicide prevention practices is given, but without taken the position/role of the agent in the field into account. So eventual differences in vision due to power relation, or role are not articulated.

Line 71 suggest that community based efforts have a unified approach…. In my opinion, that is a rather odd perspective … so how is community based defined: does it regard local participants (clients) of the intervention, does it regard local agents involved in the implementation of the unified approach or does it mean that communities should be asked to co-design and collaborate in the implementation??

In line 83 you suggest that no studies have explored how community based preventions in Australia are defined… but there is a policy “suicide prevention Australia” does that document not go into what community based interventions are? How can it specify guidelines??

Method: the research is described as qualitative. Is it in the ethnographic tradition? Often, in qualitative empirical research the (social) context of the participants involved is of importance, since this context affect their perspective taken. In this research the roles of participants in the domain of community based interventions is not taken into account. Participant do differ in their role, in this research, but this role in not made explicit, not in the purposive sampling (inclusion/exclusion criteria and the number of people in a particular role) nor in the qualitative analyses of the results.

Regarding the semi-structured interview/focus group, a topic list is not given, only the opening question. So how is the semi-structured interview/focus group conducted. If it is only an opening question, it is rather an unstructured interview.. but given the rather specific research questions: what are the community based practices, and the corresponding prompts: who is involved/ what is their role in community, what are their experiences with suicide, what are their experiences with implementing suicide prevention programs, what is their domain specific knowledge, their knowledge about managing/implementing projects, what assets/resources do they use, etc. an unstructured interview is seems not appropriate (since experiences and knowledge about such interventions are of interest).

In the section of procedure, aspects of data analyses are discussed

Results.

The results are just a summary of the qualitative data. There are some themes distinguished, but how these themes are related to one another is not discussed. Some themes are rather confusing . for example in elaboration on Knowing ‘community’ the description of community is discussed (although it is not made clear that ‘community’ can have different roles in community based interventions, such as informant about needs and resources, co-designer of the intervention, , implementer of a prescribed intervention, collaborator in the implementation) as well as what a community based intervention is. So within a section you switch from an agent (the community) to a ‘product’ (the intervention).

I advice, a more explicit research question and objective, that also refers to the literature about community engagement, community ownership and/or social innovations (for example). This literature provides, together with the “suicide prevention Australia” policy a lens to interpret the qualitative results. The qualitative analysis can still be thematic (and data driven), but the lens provides a perspective to get a deeper understanding of what is going on in the field of community based suicide prevention, so more specific guidelines and further research can be formulated.

For additional comments, I refer to the comments in the PDF document.

Author Response

please see the attached PDF with the PDF comments addressed, and table at the end of this document with additional point-by-point responses to reviewers comments

Reviewer 3 Report

Line No.

41 - needs editing.  Use being instead of are.

63 -change: ... or have not accessed ...

68 - might be good to list the authors.

110 - ... or in a research or policy development role ...

203 capitalized It's

References: Titles of journals, sources of articles, and volume numbers should be in italics for better clarification.

545 - When there are multiple pages, use pp.

558 - Most writing formats (e.g., APA) require listing all authors first in citations (up to six) before using et al.  In references, list up to six authors instead of et al.

Round 2

Reviewer 2 Report

Second review Grattidge et al, Defining Community-Based Suicide Prevention in the Context of Rural Australia: A Qualitative Study

The paper has improved much.

But still I have some comments

Main comments

Still, the research problem is not formulated well; Why is it important to know what people in the domain of suicide prevention regard as community based programs?? It is too simple to just refer to policy makers and local professionals involved. Do those policymakers and expert feel a problem about community based programs. So what is problematic for them about community based programs??  The different objectives of stakeholders with different role could be illustrated .

Moreover, in terms of evaluation research, what is the focus of this research? Is it about exploring boundaries and opportunities to design an effective intervention? Is it about monitoring the implementation of the intervention (I don’t think so, because interventions themselves are not discussed) or is it about testing the effectiveness of interventions (you could look at Bowen, D. J., Kreuter, M., Spring, B., Cofta-Woerpel, L., Linnan, L., Weiner, D., ... & Fernandez, M. (2009). How we design feasibility studies. American journal of preventive medicine36(5), 452-457. To describe the kind of study you are conducting .

The research objective needs more elaboration and should be in line with the research problem defined .

Method: You state that the research is in an ethnographic tradition. So context and role position of the informants matter. In the quotes the role of the participant is indicated but the effect different roles could have on defining community based interventions is not elaborated on. So why is an ethnographic approach suitable (or did you use a qualitative research in a more positivistic tradition in which context and roles do not matter since you are looking for general knowledge..

The population of interest of this study is not well defined (what do you mean with working experience with.. in the result suction it is sometimes suggested that it regards only people with paid jobs, professionals, experts. ) Moreover elaborate to what extent a sample from this population can give a valid and reliable answer to the research question. And provide sufficient information concerning the objective of the study.

Results.

The results are just still summary of the qualitative data. There are some themes distinguished, but how these themes are related to one another is not discussed. Please make an explicit distinction between the actors (community, in terms of the group of people the interventions is provided to versus provided by), and the intervention (adjusting the innovation to local traditions/beliefs, versus engagement in implementation of local community members as providers of the intervention and their different roles in the engagement of the implementation: informing community members, identifying community members at risk, providing support to those at risk)

Conclusions

Please elaborate on the academic and societal merit of this study.

For additional comments, I refer to the comments in the PDF document.
